# MagicFight: Personalized Martial Arts Combat Video Generation

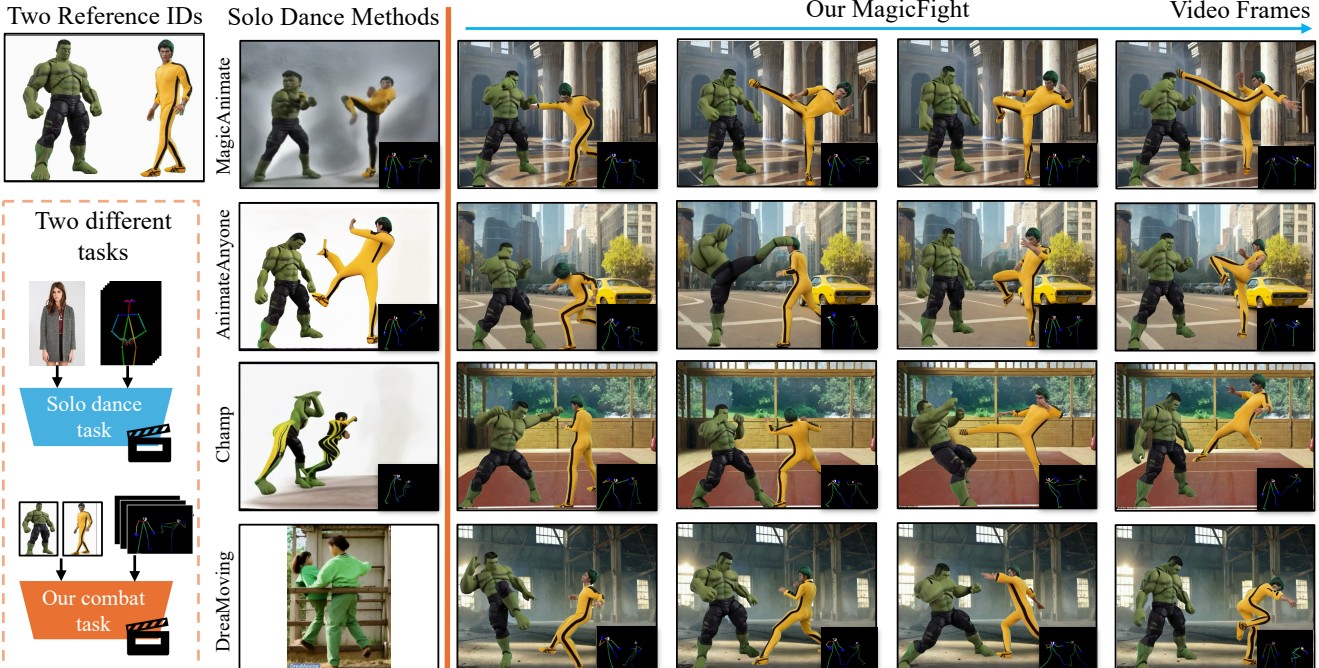

**Figure 1: Our method is the first method capable of generating high-quality martial arts combat videos. It takes two reference ID images and a conditioned pose sequence as input and generates a video that maintains consistency in both IDs and action. The solo dance methods struggle with this new task. The right part showcases our results.**

## ABSTRACT

Amid the surge in generic text-to-video generation, the field of personalized human video generation has witnessed notable advancements, primarily concentrated on single-person scenarios. However, to our knowledge, the domain of two-person interactions, particularly in the context of martial arts combat, remains uncharted. We identify a significant gap: existing models for single-person dancing generation prove insufficient for capturing the subtleties and complexities of two engaged fighters, resulting in challenges such as identity confusion, anomalous limbs, and action mismatches. To address this, we introduce a pioneering new task, Personalized Martial Arts Combat Video Generation. Our approach, MagicFight, is specifically crafted to overcome these hurdles. Given this pioneering task, we face a lack of appropriate datasets. Thus, we generate a bespoke dataset using the game physics engine Unity,

meticulously crafting a multitude of 3D characters, martial arts moves, and scenes designed to represent the diversity of combat. MagicFight refines and adapts existing models and strategies to generate high-fidelity two-person combat videos that maintain individual identities and ensure seamless, coherent action sequences, thereby laying the groundwork for future innovations in the realm of interactive video content creation.

## CCS CONCEPTS

• **Computing methodologies → Image and video acquisition**.

## KEYWORDS

Video Generation, Multi-Modal Generation, Diffusion Model, AIGC

## 1 INTRODUCTION

Video generation has emerged as a prominent field in AI research in recent years, with the creation of personalized videos representing a subtask of significant commercial and artistic value. When the specified subject is a human, this process, also known as character animation generation, entails providing an image of the source character, whereupon the model generates a realistic video following a sequence of poses specified by the user. This task boasts many potential applications, including online retail, entertainment

videos, art creation, and virtual characters, among others. Numerous studies have explored image animation and pose transfer by GAN[6, 34, 37–39, 50, 53, 56], serving as foundational work.

In recent years, diffusion models[14] have demonstrated their superiority in text-to-image [2, 19, 29, 33, 35, 36] and video generation [5, 9, 11, 13, 15, 16, 22, 31, 40, 43, 46]. Numerous researchers have utilized the architecture of diffusion models to explore video generation conditioned on given image [8, 11, 43, 55]. However, when applied to human animation, for which they are not specifically designed, these methods often produce character appearances that do not match the original image, leading to videos that lack movement coherence. For fashion video generation, DreamPose[21] introduces an adapter to fuse CLIP[32] image features into Stable Diffusion [35] and finetunes on the input sample.

Recent works specializing in human dance video generation, including DisCo [42], MagicAnimate [47], AnimateAnyone [18], MagicDance [7], DreaMoving [10] and Champ [57] exhibit similar approaches and network structures. DisCo [42] extracts character and background features via ControlNet[52] while it shows serious flaws in generating the ID. Other methods [7, 10, 18, 47, 57] all aim to solve the issue of ID appearance. Each employs its own appearance encoder, utilizing a parameter-rich encoder like ControlNet for multi-scale and detailed ID feature extraction from the original image. They design an effective pose guide for controllability and a temporal module for smooth interframe transitions. Furthermore, the pivotal element is the training data they have amassed. By leveraging large-scale, high-quality datasets, these methods can animate arbitrary characters.

However, all the aforementioned methods fall short in human fighting video generation involving multiple subjects. As these methods are designed for single-person dancing, they accept a single ID and a single-person pose sequence, and their training datasets predominantly contain single-person dance videos such as TikTok [20]. Besides, the absence of network design for multi-person and the lack of multi-person dataset preclude these existing works from effectively generating multi-person fighting videos. Hence, we introduce a new task: personalized martial arts combat video generation. There are three primary distinctions between our new task and the existing ones: 1) Subject number: The existing task focuses on solo dances, whereas ours involves two individuals. 2) Motion type: While fashion and dance videos emphasize slow and individual movements, martial arts combat requires capturing complex kung fu and varied poses. 3) Interaction dynamics: Unlike solo dance with no interactive dynamics, martial arts combat necessitates depicting the intricate interplay between two-person, highlighting the authenticity of the generated video.

In this paper, we design a base method MagicFight for our proposed new task named personalized martial arts combat video generation. To establish this foundational method, we address existing issues in current techniques and investigate dataset production, processing, and training strategies. Our main contributions include:

(1) For the first time, we delineate two-person fighting from one-person dancing. We create a dataset of martial arts combat videos named KungFu-Fiesta (KFF) and establish data cleaning rules for dataset quality and diversity, laying a solid foundation for this new task.

(2) We introduce a multi-modal personalized network to learn conditioning on two reference IDs, pose, background, and prompt, focusing on the dynamic complexities of combat. With the personalized attention layer (ID-attn), we address the clothing and body misattribution problem in our task.

(3) In the inference stage, we introduce body-shape adaptive strategy to automatically adjust the preset pose map, aligning the generated video more closely with the expected body shape. For arbitrary long video generation, we use a clip fusion technique to ensure continuity between clips.

(4) We conduct a comprehensive ablation study from both the dataset and model training perspectives. We explore the properties, size, and quality requirements for our dataset on this task, and offer insights and guidance about the effect of different training components on overall performance. We creatively propose the Mixture Data Finetuning strategy, which mixes self-made two-person fashion videos and KFF dataset for training, in order to take full advantage of different data domain.

## 2 RELATED WORK

### 2.1 Conditional Video Generation

The field of video generation has advanced significantly, thanks to diffusion models adapted from text-to-image (T2I) techniques. Research efforts [9, 15, 16, 22, 26, 31, 40, 46, 49] introduce frame attention and embedding temporal layers within T2I models. Initiatives like Video LDM [5] advocate for image pretraining before engaging in video temporal training, and AnimateDiff [11] brings motion modules to T2I models without the need for specialized adaptation. Expanding into image-to-video transformation, VideoComposer [43] stands out by integrating images as conditional inputs during training. VideoCrafter [8] distinguishes itself by melding text and visual features from CLIP into its cross-attention mechanism. The Stable Video Diffusion (SVD) [4] signifies a quantum leap in enhancing video quality and dynamic representation. With W.A.L.T [12] pioneering through its VAE Encoder in choosing optimal latent representations, and the Sora [1] setting new standards for high-definition, realistic video outputs, these advancements mark a decisive turn towards refined, high-quality video generation.

### 2.2 Human Video Generation

Recent studies highlight the incorporation image-to-video diffusion model into human video generation. PIDM[3] introduces texture diffusion blocks to infuse desired texture patterns into the denoising process for human pose migration. LFDM[28] synthesizes optical flow sequences in latent space, distorting the input image based on specified conditions. LEO[44] represents motion through a series of flow maps, using a diffusion model to synthesize the motion sequence. DreamPose[21] utilizes a pre-trained stable diffusion model, introducing an adapter to model the image embeddings extracted by CLIP and VAE. DisCo[42], inspired by ControlNet, decouples pose and background control. MagicAnimate, AniamteAnyone, and Champ build primarily on the DisCo and advance the improvement of appearance alignment and motion control mechanisms. However, these methods still struggle with issues like ID appearance inconsistency and temporal instability. Moreover, no method yet exists for generating martial arts combat videos or focusing on two-person motion video.

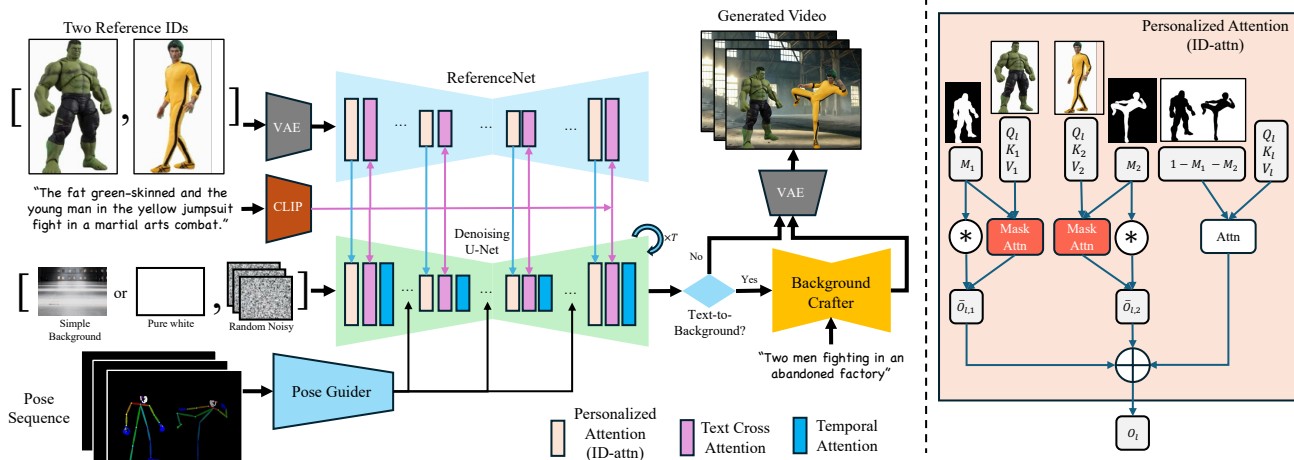

Figure 2: Our MagicFight has 4 conditions for combat video generation, two reference IDs, a text prompt, a background image, and pose maps. The action of each frame is controlled by Pose Guider. The two IDs are personalized by our Personalized Attention (ID-attn) layer which can generate the respective appearance to the desired place. The user can provide a simple background image or use a pure white background and then generate a complex and reasonable background by our Background Crafter. With the long video generation technique, we can make arbitrary long videos (typically 10 seconds in our test).

Table 1: Details of Our Martial Arts Combat Video Dataset

| Scene Category | Island | Rainforest | Palace | City | Mountains | Snowfield |
|---|---|---|---|---|---|---|
| Number of Videos | 42 | 38 | 45 | 41 | 39 | 45 |
| | Desert | Seaside | Bridge | Open Field | School | Boxing Gym |
| Number of Videos | 43 | 40 | 37 | 44 | 36 | 35 |
| Action Category | Boxing | Kicking | Wrestling | Jeet Kune Do | Somersault | Weapon Combat |
| Number of Videos | 40 | 42 | 43 | 39 | 38 | 44 |
| | Rapid Attack | Blocking | Dodging | Judo | Wing Chun | Finishing Move |
| Number of Videos | 37 | 45 | 36 | 42 | 41 | 34 |
| Character Category | Male Staff | Female Staff | Fat Person | Thin Person | Tall Person | Short Person |
| Number of Videos | 43 | 41 | 38 | 42 | 37 | 39 |
| | Beauty | Soldier | Student | Elderly | Athlete | Martial Artist |
| Number of Videos | 40 | 45 | 34 | 36 | 42 | 44 |

## 3 METHODS

First, we analyze the existing problems in Sec. 3.1. Then, we detail our dataset creation process in Sec. 3.2, and our model architecture in Sec. 3.3. We describe the training and inference in Sec. 3.4 and 3.5, respectively.

### 3.1 Existing Problems and Motivation

We commence with an analysis of the challenges that existing models face in generating scenes with complex character interactions as shown in Fig. 1. 1) During the generation with two-person interactions, a common issue is misattribution of clothing and body parts, particularly when characters are close. For instance, the woman's left leg in the short skirt might be incorrectly merged with the man's pants, with color inaccuracies also occurring. These issues highlight the necessity for a customed two-person model to address the misattribution problem. 2) Besides, missing body parts also frequently occur, like duplicated legs or absent feet. This issue stems from the inadequate data on leg lifting and kicking actions in the human dance dataset and the absence of foot keypoints in the pose maps, leading to the model's poor perception of leg and foot features, which is thirsty for a tailored martial arts dataset. 3) Moreover, existing models often produce medium-sized characters, overlooking the diversity in body shape. For instance, muscular "Hulk" and bony people are frequently underrepresented. Hence, we aim to solve the problem of mismatch between body type and

given pose, adapting to any body shape during inference. Our research seeks to mitigate the aforementioned problems and lay the groundwork for future endeavors.

### 3.2 KungFu-Fiesta Dataset Creation

This section details our first martial arts combat video dataset named KungFu-Fiesta (short for KFF). We make 4 scenarios for this dataset creation and finally chose the Unity scenario, details about it are in our appendix. With Unity, an advanced game physics engine, it can create highly realistic 3D character models and action animation in a simulated world, and by rendering the scene from an angle and exporting them to video, it is possible to create a large number of highly realistic martial arts combat videos. For the diversity and complexity of the dataset, we design hundreds of character IDs with different identities, covering more than 100 kinds of paired fighting actions, and a variety of shooting angles in 20 different scenes. After careful design and production, we capture more than 500 high-quality videos. Each video sample is about 10 seconds with 60 fps, ensuring the coherence of the action. In KungFu-Fiesta, each sample contains a combat video, two reference images of character IDs (for short reference IDs), and a pose map sequence, providing researchers with more conditions. The details of the dataset are shown in Table 1.

### 3.3 Multi-Modal Personalized Network

Our model is an extension of the Stable Diffusion (SD), so we inherit its VAE [23], denoising UNet and CLIP encoder. Fig. 2 provides an overview of our framework. The input to the network is multi-frame noisy latent $z_t \in \mathbb{R}^{F \times c \times h \times w}$ (timestep $t$). In order to utilize the general knowledge of human motion, our model is based on the pretrained model of AnimateAnyone [18] which is for single-person dancing video generation. The framework consists of three key components: 1) ReferenceNet is responsible for encoding the appearance of the two reference IDs; 2) Pose Guider is for controlling the two-person's fight by pose map; 3) Temporal layer, the

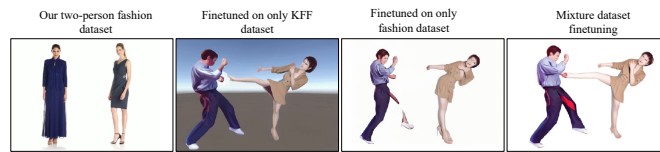

Figure 3: Dataset ablation study. We mix the KFF dataset with our remade UBC fashion dataset (two videos spliced into a two-person video) for training, which improves the clarity and quality of the video. Training with the fashion dataset alone could not generate some martial arts movements, such as kicks, as the movements in this dataset are too simple.

attention layer between these frames is to ensure the continuity of the character's movement. AnimateAnyone proposes to use a lightweight pose controller with only 4 simple convolutional layers since the pose control of single-person is easy. However, our two-person martial arts situation is more complex and the input pose becomes two-person. We finally choose to use a large Pose Guider like ControlNet.

**Personalized Attention Layer.** In our task, the given reference IDs provide detailed appearance information. However, the ReferenceNet of AnimateAnyone is designed and trained for single-person feature extraction. Thus, we feed 2 ID images into ReferenceNet alongside the batchsize dimension to extract their features $[r_1, r_2]$, and then they are fed into the denoising U-Net. As shown in Fig. 2, our personalized attention (ID-attn) layer replaces the original self-attention layer of SD. Given the feature map $x_l \in \mathbb{R}^{F \times h \times w \times c}$ in the $l$-th ID-attn layer and ID features $r_1, r_2 \in \mathbb{R}^{h \times w \times c}$, ID-attn is performed as:

$$
\begin{aligned}
\bar{O}_{l,i} &= \text{MaskAttn}(Q_l, K_i, M_i)V_i, \\
O_l &= M_1\bar{O}_{l,1} + M_2\bar{O}_{l,2} + (1 - M_1 - M_2)\text{Attn}(Q_l, K_l)V_l,
\end{aligned}
\tag{1}
$$

where $Q_l$ denotes the Query from $x_l$, $[K_i, V_i]$ represents the Key/Value from $i$-th ID $r_i$, and $[K_l, V_l]$ denotes the Key/Value from $x_l$ itself. Computing attention between the whole $x_l$ and $r_i$ may lead to reference disruption. Thus, MaskAttn() means only to keep the attention of the $M_i$ region and mask the other regions with no attention. $M_i$ denotes the target position of ID $i$, computed by the bounding box of the pose of ID $i$. So the target part of $M_i$ is from $r_i$ and the background part $1 - M_1 - M_2$ is not affected by IDs.

**Conditioned Background.** For conditioned background (pure white also OK), we concat the given background image latent with $z_t$ at channels and input to U-Net. The user can 1) provide a simple background image for end-to-end background customisation, and 2) if the user does not want to provide a background image, the conditioned background will be set to pure white, and then user can provide text prompt in Background Crafter to generate the background. Our Background Crafter is based on SDXL-Inpainting [30]. Its conditions are the original foreground image, background mask (it is easy to obtain a mask due to the white background), and text prompt. We finetune it on our dataset and follow [25] to maintain inter-frame background consistency, which is detailed in our appendix.

## 3.4 Multi-Stage and Mixture Dataset Finetuning

### 3.4.1 *Mixture Dataset Finetuning.* We propose to use a mixture of KFF dataset with our recreated two-person fashion video dataset

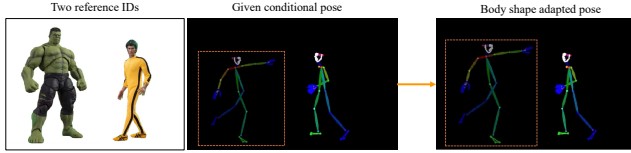

Figure 4: The body size adaptive strategy during inference.

made based on the UBC dataset [51]. It is worth noting that the UBC dataset originally contains videos of a single person walking down a fashion runway, and by splicing two randomly selected videos left and right together, we create a video dataset that simulates a two-person fashion runway, which has the advantages of a pure white and clean background, real people in the subjects, and high-definition clothing textures, and the disadvantages that the two subjects are randomly spliced together, and lack of multi-subject interactions. Based on the benefits of the two-person fashion dataset, we hypothesise that a strategy of training with a mixture of two-person fight videos and two-person fashion videos would improve the consistency and aesthetics of the appearance, thus demonstrating stronger generalisation capabilities when dealing with complex character interaction scenarios.

### 3.4.2 *Multi-Stage Finetuning.* Since our MagicFight model is finetuned on the pretrained Moore-AnimateAnyone [24][1], we propose the 2 stages finetuning. The first stage during finetuning is a spatial learning stage, using individual video frames from our KFF dataset as image input. In denoising U-Net, all temporal layers are frozen and become intra-frame attention, and the model takes the noisy single frame as input, along with the reference IDs and the pose map. ReferenceNet and Pose Guider are trained to learn the spatial distribution of the two-person fighting. The pretrained weights on the human dancing dataset are used to initialize our denoising U-Net, ReferenceNet, and Pose Guider, which adopts a ControlNet-like structure rather than the lightweight controllers in Animate Anyone. The second stage is for temporal layer finetuning, whose input is 20 frames of video clip from our KFF dataset, and network parameters except temporal layers are frozen to learn the general law of two-person fighting action.

## 3.5 Inference Strategy

### 3.5.1 *Long Video Generation Technique.* Previous diffusion-based video generation typically focuses on short video clips. For generating an arbitrary long combat video, we introduce a clip fusion technique to ensure continuity of details between clips. Specifically, we retain the $x_t$ of the last 4 frames of each clip in sampling steps. When inferring the next video clip, we use the saved 4 frames and the following 20 frames as $x_t$. During each sampling step, we superimpose the $x_t$ of the last 4 frames onto those of the first 4 frames of the currently generated clip to generate videos of arbitrary length maintaining consistency.

### 3.5.2 *Body-Shape Adaptive Strategy.* In the video generation process, considering the possible differences in body types (e.g., height, body shape, etc.) between the given pose map and the reference IDs, we face a challenge to ensure the body shape in the generated video is consistent with those in the reference IDs. For example, if the reference ID is a tall and chubby person, and the given pose

---

[1]Since AnimateAnyone has no released code, we use reproduction version.

**Table 2: Quantitative comparison of the KFF reconstruction benchmark.**

| Method | SSIM ↑ | PSNR ↑ | LPIPS ↓ | FVD ↓ |
|---|---|---|---|---|
| MagicAnimate | 0.888 | 22.479 | 0.090 | 623.00 |
| AnimateAnyone | 0.873 | 21.398 | 0.087 | 572.22 |
| Champ | 0.877 | 22.018 | 0.066 | 523.01 |
| Ours | **0.893** | **23.756** | **0.058** | **454.62** |

is from a little girl, it may result in visual incongruity. Thus, we introduce a body-shape adaptive strategy, which is shown in Fig. 4. First, we predict the pose of the reference IDs, and compute the center of mass of the character's keypoints in the horizontal (x-axis) and vertical (y-axis) directions. Similarly, we also compute those of the given conditioned pose map. Subsequently, we compute the body scale factors in the x/y-axis. With these scale factors, we scale the coordinates of all the key points in the pose to ensure that the generated video content meets the action requirements and is faithful to the body shape of the reference IDs.

## 4 EXPERIMENT

### 4.1 Implementation

To validate MagicFight's efficacy in generating martial arts combat videos with diverse IDs, we make two benchmarks, KFF reconstruction benchmark and open-set combat generation benchmark, to evaluate our model. We employ pretrained DWPose to estimate pose maps, including body, hands, and foot. All finetuning experiments are conducted on 8 NVIDIA A6000 GPUs, each with 48G GPU memory. In the first finetuning stage, we sample individual frames at a frame interval of 6, then adjust the frames to a resolution of 704×512. Finetuning is performed for 20,000 steps, with a batch-size of 2 per GPU. In the second finetuning stage, we finetune the temporal layer for 10,000 steps with a video sequence of 20 frames, frame interval of 6, and batchsize of 2. Both learning rates are set to 2e-6. During inference, we employ the DDIM sampling for 25 denoising steps. We adopt our long video generation technique and body-shape adaptive strategy for better generation. For comparison with human dance generation methods, we test all methods on the same benchmark, detailed in Sec. 4.2.

### 4.2 Qualitative and Quantitative Evaluation

Figs. 5 and 6 illustrate our method's capability to produce controllable combat videos for various character types, such as real, cartoon, robotic, and humanoid. Our method produces high-definition videos with realistic character details. It ensures temporal consistency with the reference IDs and maintains continuity between frames, despite significant motion.

To illustrate our method's superiority over other video generation methods, we assess them on two bespoke benchmarks: KFF reconstruction benchmark and open-set combat generation benchmark. For quantitative evaluation of the reconstructed video quality, we utilize metrics such as SSIM[45], PSNR[17], FVD[41], and LPIPS[54]. The evaluation of the open-set video generation benchmark incorporates user ratings, FVD[41], and NIQE metrics[27]. In our experiments, we follow the computation of FVD as VideoGPT [48].

Given that SSIM and PSNR may not match human perception, we employ LPIPS and NIQE as complementary evaluation metrics.

**Table 3: Quantitative comparison of open-set combat generation benchmark.**

| Method | FVD ↓ | NIQE ↓ | User Score ↑ |
|---|---|---|---|
| MagicAnimate | 937.34 | 5.23 | 2.05 |
| AnimateAnyone | 1178.57 | 4.68 | 3.77 |
| Champ | 1130.22 | 4.56 | 3.89 |
| DreaMoving | 1851.93 | 5.92 | 0.41 |
| Ours | **812.77** | **4.14** | **4.12** |

LPIPS quantifies perceptual similarity, offering a closer representation of the human eye's subjective judgment. NIQE acts as a reference-free image quality evaluation metric tailored to appraise the visual aesthetic quality of images. A user study is conducted to evaluate the subjective quality comprehensively. Forty users review the results from all methods. Each sample consists of IDs image, pose sequence, text prompt, and results from each method. Participants rate the quality of each video on a scale from 1 to 5. The evaluation primarily focuses on the IDs' similarity, pose control and visual appeal. We calculate the average scores for each method and gauge potential popularity and practical value.

*4.2.1 The KFF Reconstruction Benchmark.* KFF reconstruction involves generating a reconstructed video given two reference IDs and the pose sequence. Our KFF reconstruction benchmark comprises 100 video clips, each with around 180 frames. The selection criteria for this benchmark require the test video's character and action to match the training set's domain, yet not exactly existing in the training set. Quantitative comparisons are detailed in Table 2, where our results significantly surpass those of other methods, particularly in the reconstruction metrics. Qualitative comparisons are displayed in Fig. 5. We employ the web demo or the code of the compared methods. These methods can't provide conditional backgrounds or generate new backgrounds, and DreaMoving is a vertical screen resolution (so we keep it vertical). Refining human details demands high precision, while our method maintains detail consistency.

*4.2.2 Open-Set Combat Generation Benchmark.* The open-set combat generation benchmark focuses on the open world of human interactions video. We collect 20 IDs from the game community and the Internet, comprising 40 test samples. We generate about 10 seconds of video for each sample. The selection criteria for this benchmark allow characters, actions, and backgrounds to span any data domain, with no restrictions on data source. Our approach yields the best quantitative results, as shown in Table 3. DisCo, AnimateAnyone, and MagicAnimate undergo extensive pre-training on human image datasets, learning basic single-person patterns, thus lacking multi-person interaction knowledge. In contrast, our mixture dataset training on the KFF and two-person fashion dataset yields superior results compared to these methods. Our method demonstrates that without explicit segmentation, the model can discern foreground-background relationships from multi-subject movements. Furthermore, our model excels at maintaining visual continuity in complex action sequences, demonstrating robustness in handling varied character appearances.

### 4.3 Ablation Study

*4.3.1 Dataset Attributes.* To elucidate the differences in dataset attributes and explore their impact on finetuning efficacy, we focus

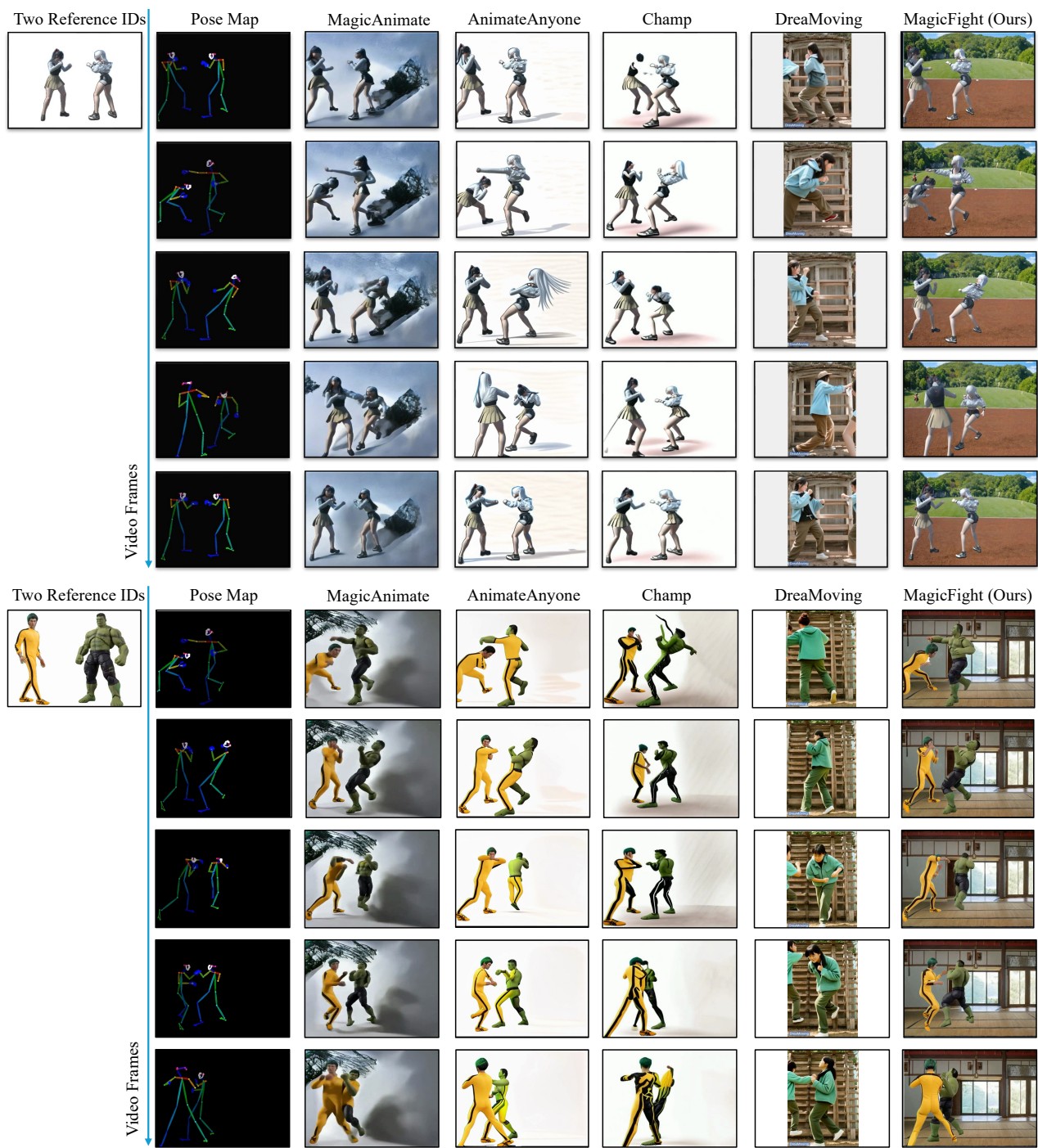

**Figure 5: The results on two benchmarks. These solo dance models exhibit missing body parts and wrong actions, and they cannot be conditioned on background or generate background by prompt. Our MagicFight significantly mitigates these issues.**

on the data scale, number of character IDs, actions, backgrounds, and the mixture with two-person fashion data.

**Data Scale.** Data scale is a key factor in evaluating the fine-tuning effectiveness. Theoretically, a larger dataset is believed to provide richer information for training, enhancing the model's generalization to new scenarios. Table 4 indicates that as the data scale

increases, the model shows improvement in FVD and user scores exhibiting superior visual quality.

**Number of Character IDs.** Among the attributes, the number of character IDs is a crucial factor under the assumption that more IDs offer diverse learning opportunities for character traits, thereby enhancing video diversity and realism. As depicted in Fig. 7,

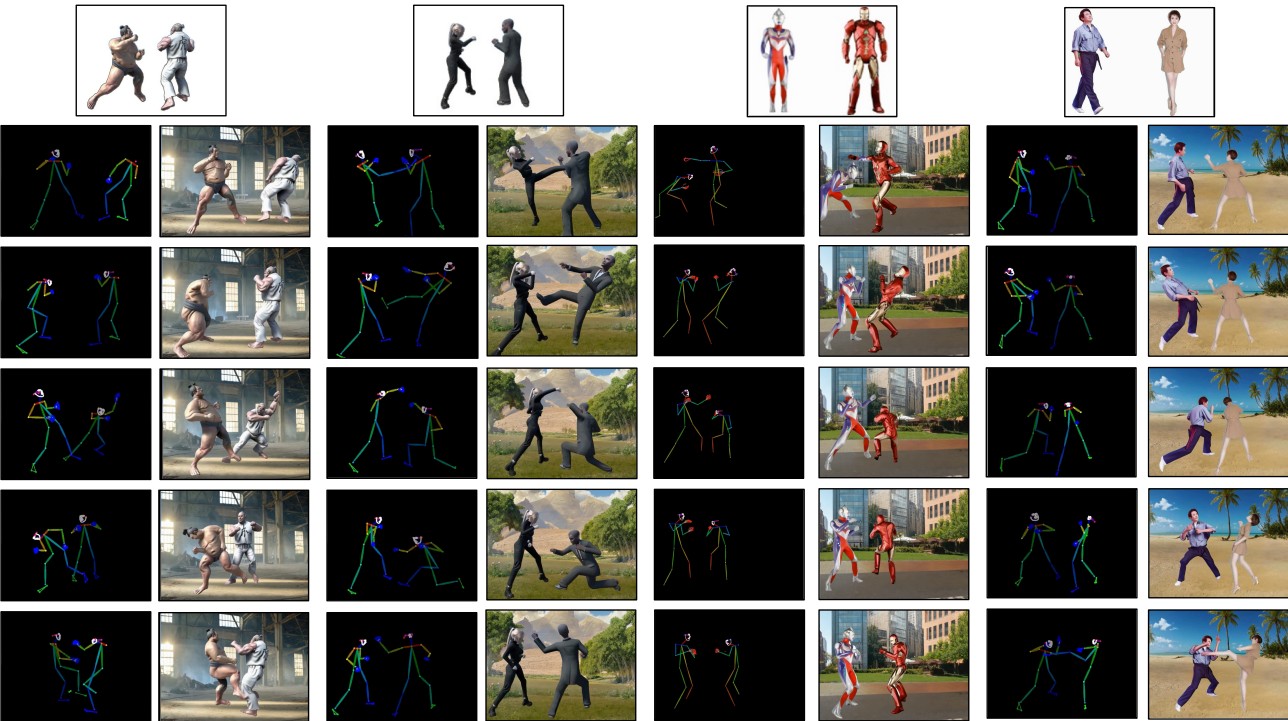

Figure 6: The MagicFight results in open-set combat generation with smooth movements and consistent IDs. Because of page limits, we give more results in our appendix.

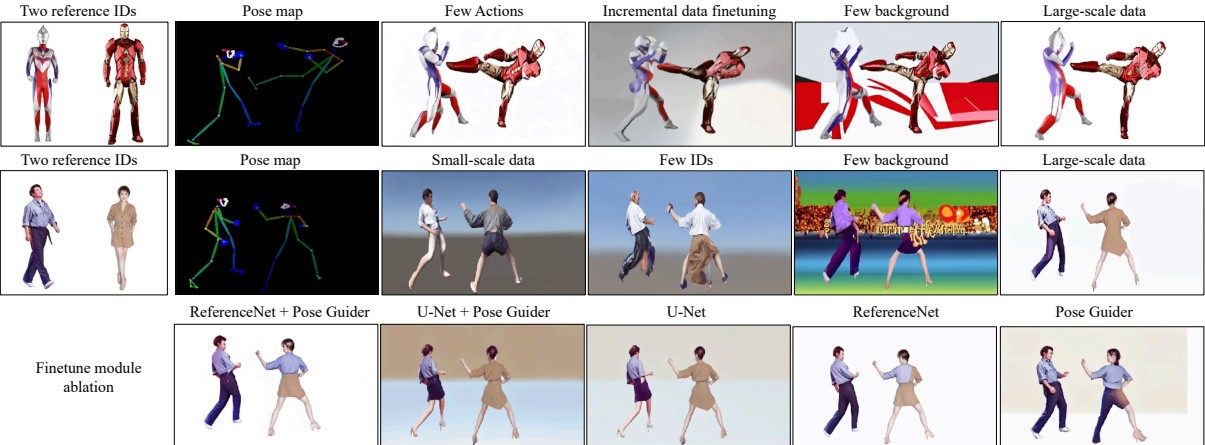

Figure 7: Ablation Study. 1) ID appearance is ensured with sufficient IDs in the training data. 2) Adequate action in the training set is helpful. 3) Currently, end-to-end way struggles to handle complex backgrounds.

the number of IDs significantly impacts the training effectiveness more than other attributes. Insufficient character IDs can lead to overfitting to specific characters. Quantitative results in Table 4 demonstrate our dataset's superiority across attributes. These results validate our hypothesis highlighting prioritizing character diversity in dataset construction.

**Number of Action.** We hypothesize more actions in our dataset should help the dynamics and complexity of martial arts videos. The result presented in Table 4 and Fig. 7 shows that an increase in action types somewhat improves video quality metrics, though not as significantly as with IDs, suggesting that ID diversity is

more crucial than action variety. Qualitative analysis reveals that more actions yield videos with complex interactions like overkick. Therefore, for open-set actions, the dataset should be constructed to include as many diverse martial arts types as possible.

**Mixture Dataset Finetuning.** We explored the impact of using the mixture video dataset. Specifically, we compare two training strategies: 1) training on our KFF dataset alone, and 2) training by mixing KFF with our remade two-person fashion video dataset based on UBC [51]. It is worth noting that the UBC dataset only contains single person walking in a fashion show. By combining two videos side by side, we create a new dataset that simulates a

**Table 4: Quantitative Comparison of Ablation Study**

| Setting | Number of Videos | Number of IDs | Number of Actions | Number of Backgrounds | FVD ↓ | User Score ↑ |
|---|---|---|---|---|---|---|
| Small-scale | 10 | 16 | 24 | 2 | 948.45 | 3.29 |
| Medium-scale | 50 | 60 | 80 | 4 | 880.68 | 3.98 |
| Large-scale | 160 | 180 | 120 | 4 | **812.77** | **4.25** |
| Few IDs | 40 | 8 | 80 | 2 | 1012.83 | 2.56 |
| Many IDs | 40 | 70 | 80 | 2 | **867.43** | **4.11** |
| Few Actions | 40 | 30 | 24 | 2 | 885.62 | 3.47 |
| Many Actions | 40 | 30 | 80 | 2 | **848.14** | **4.09** |
| Single Background | 100 | 50 | 70 | 2 | **816.46** | **4.23** |
| Various Backgrounds | 100 | 50 | 70 | 12 | 923.19 | 2.93 |
| Freeze Denoising U-Net | 160 | 180 | 120 | 4 | 908.64 | 3.22 |
| Freeze ReferenceNet | 160 | 180 | 120 | 4 | 838.14 | 3.91 |
| Finetune Pose Guider Only | 160 | 180 | 120 | 4 | 925.83 | 2.73 |
| Finetune ReferenceNet Only | 160 | 180 | 120 | 4 | 913.92 | 2.84 |
| Finetune Denoising U-Net Only | 160 | 180 | 120 | 4 | 823.91 | 4.01 |
| Finetune Temporal Layer Only | 160 | 180 | 120 | 4 | 846.70 | 3.57 |
| Train the Entire Network | 160 | 180 | 120 | 4 | **812.77** | **4.25** |
| Incremental Data Finetuning | 50+110 | 60+120 | 80+40 | 4 | 821.07 | 4.11 |
| Full Data Finetuning | 160 | 180 | 120 | 4 | **812.77** | **4.25** |
| Only KFF Dataset | 160 | 180 | 120 | 4 | 812.77 | 4.25 |
| Mixture Dataset | 600 | 500 | 400 | 10 | **756.43** | **4.78** |

two-person fashion walk with 3 benefits: 1) pure white and clean background, 2) real people, and 3) high-definition clothing textures. As shown in Fig. 3, the result shows that mixture dataset finetuning significantly improves the clarity and texture aesthetics compared to training with KFF alone. While KFF emphasizes intense fighting, the two-person fashion videos demonstrate calm and clear portraits and this diversity leads to a comprehensive and flexible understanding of character appearance and movement. However, training with only the fashion dataset could not render some martial arts actions, such as kicking, as this dataset has only simple actions.

*4.3.2 Finetuning Strategies.* We maintain the same training set for each experiment. For finetuning module ablation, we analyze denoising U-Net, ReferenceNet, Pose Guider, and temporal layers. Besides, we analyze the impact of incremental data finetuning.

**Finetuning Module Ablation.** Module-specific finetuning targets for the optimization of specific parameters while retaining the most original generative capabilities. We hypothesize that finetuning different modules has different effects. Table 4 and Fig. 7 present results of differences in finetuning modules, leading us to the following preliminary conclusions: 1) Without finetuning the denoising U-Net, denoising loss can only be reduced to around 0.4 but not further to 0.2. 2) Untrained ReferenceNet or Pose Guider leads to body distortions, missing parts, or inconsistent IDs. 3) Although the first stage of finetuning may yield suboptimal results, performance can be significantly improved in the second stage. 4) Finetuning solely the temporal layers often causes artifacts, distorted body, and background anomalies in certain samples.

**Incremental Data Finetuning.** We initially finetune with medium-sized data and, after every 10,000 steps, gradually introduce new data. The results reveal that its impact on enhancing diversity and realism is negative. We hypothesize that gradually increasing the data scale may lead to a suboptimal model weight.

## 5 LIMITATIONS AND FUTURE DIRECTIONS

This part discusses the limitations of our proposed methodology and outlines directions for future research. Our approach has the following limitations: Firstly, like many visual generative models, ours struggles with perfect foot and hand generation. Secondly, our reference IDs offer only a single-angle view, making the generation of occluded parts during action problematic; for instance, if the reference image lacks a frontal view, the generated facial quality is poor. Thirdly, when the two people overlap for some complex action like wrestling, pose control becomes chaotic.

Then we introduce our future work. First, when handling complex dresses, like cartoon costumes or clothes with ribbons, our method may exacerbate flash frame issues. We suggest manually labeling the pose map. Secondly, background control remains a significant challenge. The existing framework cannot generate backgrounds that are dynamic (such as flowing water, fire, and rain), have complex layouts, or have passers-by. We are working hard to propose a new framework that can generate dynamic foreground and background in the same model. Finally, our current approach focuses on the case where the camera is stationary, and all of our training videos are camera-still. In order to adapt to the situation of dynamic shots in real martial arts movies (e.g., complex situations such as slow camera movement, rotation, or even switching of shots, etc.), future work will focus on introducing modeling of the camera position for the network and producing more video datasets with camera movement, which will lead to a more realistic and higher-degree-of-freedom generation of martial arts videos.

## 6 CONCLUSION

This paper introduces a foundational framework for generating martial arts combat videos, transforming two characters into combat video with pose sequences, ensuring appearance consistency and temporal stability. We make the first combat video dataset named KungFu-Fiesta (KFF), specifically designed for this task, created using the Unity engine to ensure diversity and physical realism. We finetune a multi-modal personalized network to acquire combat knowledge, aiming to preserve the intricate appearance of IDs while enabling efficient pose control and temporal continuity. The user can specify a background image or easily customize the background through the Background Crafter by text prompt.

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
