# OpenReview forum: "MagicFight: Personalized Martial Arts Combat Video Generation"
_acmmm.org/ACMMM/2024/Conference — MM2024 Poster_

### Official Review · Reviewer_WG4k · 2024-05-21

**Rating:** 2
**Confidence:** 3

**Summary:**

This article introduces a new task: personalized martial arts combat video generation. It proposes a specialized dataset constructed using Unity to support this task, which includes 3D characters, actions, and scenes.

**Strengths:**

1. This paper introduces a new task, which is interesting. The combat videos, of course, are more complex than single dance videos.
2. A new dataset should be considered as a contribution. Especially when it is a new task.
3. This work did consider some detail problems, such as body size adaptive strategy.

**Limitations:**

1. My greatest concern is about the authenticity of the results presented in the paper. I have observed that the method proposed in the paper shows abnormal rendering of light and shadow in some results, particularly in parts that appear to be 3D assets. This is especially evident in the example of the two girls fighting in the upper part of Figure 5. It is clear that the edge shadows of the input images are significant and are reflected in other methods. However, the results of the proposed method exhibit an unusually smooth and markedly different skin shading.
2. The method presented in this paper is also difficult to comprehend. I noticed that the inputs to the method are character IDs, text descriptions, background images, and pose sequences. However, additional character mask information appears in the Personalized Attention proposed in the paper. Moreover, it is determined before the VAE whether the background is generated through text. Firstly, how is this mask obtained? Can masks for different characters be estimated based on the given pose sequences for various reference IDs? Secondly, the text-generated background is confusing, with the example "two men fighting in an abandoned factory." The only useful information in this sentence is the abandoned factory. Furthermore, for the given method, it seems that it cannot be any other choice besides two men, nor any other action besides fighting, even though these are essentially useless information. Lastly, I believe the most useful part of the method is the character ID and pose. Is the text description truly meaningful?
3. The proposed task is met with skepticism in its potential applications. The task presented in the paper requires the synthesis of fighting videos using reference IDs and pose sequences. However, the information from the reference IDs is limited, as mentioned in the limitations, which can lead to issues with occluded parts. The information loss due to the modality is irremediable. Moreover, the action sequence is the most significant workload in the given application. I believe that the task of generating action sequences is more valuable than the task proposed in the paper, as the existing process can quickly render videos from action sequences and 3D assets, including more complete and detailed results, scenes, and lighting. In contrast, the task proposed in this paper does not reduce the input requirements (both need pose and character, and additionally require text) or improve the quality of the results (due to the modal disadvantage of 2D characters and the significantly inferior lighting inference capabilities compared to existing renderers).
4.  In summary, I currently have a high degree of suspicion regarding the authenticity of the results. Moreover, I believe the method is overly convoluted, making it difficult for me to capture the genuinely useful aspects. Additionally, while the new task is intuitively interesting, I think it is less effective in the application scenarios mentioned in the paper compared to other technological approaches.

**Suitability:**

2

---

### Official Review · Reviewer_Gsu9 · 2024-05-26

**Rating:** 3
**Confidence:** 2

**Summary:**

This paper proposes a controllable multi-person human-centric video generation method, MagicFight. The major contribution in this paper is a personalized attention module, which can control the appearance of different person based on the ID. Furthermore, this paper also propose a new dataset on martial art that targeted for multi-person human-centric video generation. This paper achieves strong performance on several benchmarks.

**Strengths:**

1. Multi-person video generation can be challenging, and this paper proposes an effective method towards this task.
2. New multi-person video dataset is proposed in this paper, which could potentially serve as benchmarks for future work.
3. The quantitative results are good, outperforming existing methods.

**Limitations:**

1. This paper uses mixed dataset finetuning (Sec 3.4), which could pose an unfair comparison with competing methods.
2. This paper is not the first one to work on multi-person generation. For example, [a] is an zero-shot methods, and this paper does not compare the results with [a], but only compare results with single-person generations methods (MagicAnimate, AnimateAnyone, Champ, and DreaMoving), which is unfair.
3. This paper works on **video** generation, but does not provide video comparison. Video metrics are well-known for the fluctuation; thus, only from the metrics, the visual quality and temporal smoothness cannot be well evaluated.
4. Generalization ability is not evaluated. This paper only conducts in-domain evaluation and misses out-of-distribution evaluations, such as using non-human reference images.
5. This paper only shows the results of two persons. I wonder if this method can generalize to more people?



Missing references

[a] Xu, Zhe, Kun Wei, Xu Yang, and Cheng Deng. "Do You Guys Want to Dance: Zero-Shot Compositional Human Dance Generation with Multiple Persons." *arXiv preprint arXiv:2401.13363* (2024).



Justification: my major concern lies in the incomplete experiments and lack of video result comparison.

**Suitability:**

3

---

### Official Review · Reviewer_f7vx · 2024-05-26

**Rating:** 5
**Confidence:** 3

**Summary:**

The paper introduces MagicFight, a novel framework for generating personalized martial arts combat videos, filling a gap in the field of two-person interaction video generation. Specifically, MagicFight learns from two reference IDs, pose sequences, and other conditions, employing a personalized attention layer to address misattribution issues in character generation.

Additionally, it presents the creation of KungFu-Fiesta (KFF), a bespoke dataset designed using Unity to capture the nuances of martial arts combat with a diverse range of characters and scenes, crucial for training the model.

MagicFight incorporates a body-shape adaptive strategy for inference, aligning generated videos with expected body shapes, and a clip fusion technique for generating long, coherent videos.

The paper provides a comprehensive evaluation of MagicFight, demonstrating its effectiveness in producing high-fidelity, temporally consistent combat videos and its superiority over existing methods through both qualitative and quantitative benchmarks.

**Strengths:**

1. The problem is interesting and important, i.e., the domain of two-person interactions, particularly in the context of martial arts combat, remains uncharted.

2. The motivation is clear and the proposed framework is novel.

3. Extensive experiments have been conducted. MagicFight achieves robust performance.

4. The proposed dataset will make a significant contribution to the video generation community.

**Limitations:**

Lack of User Study: Conducting a more extensive user study to gather feedback on the generated videos' realism, diversity, and user satisfaction could provide valuable insights for further refinement.

**Suitability:**

3

---

### Official Review · Reviewer_Ge7y · 2024-05-27

**Rating:** 4
**Confidence:** 3

**Summary:**

This work introduces a new method for generating personalized martial arts combat videos, addressing the complexities and challenges inherent in two-person interactions. Unlike existing models that focus on single-person scenarios such as dancing, MagicFight aims to generate high-fidelity combat videos that maintain individual identities and coherent action sequences. To support this, the authors created a bespoke dataset using the Unity game engine, ensuring a diverse range of combat scenarios.

**Strengths:**

-  This work introduces MagicFight, a new task of personalized martial arts combat video generation, filling a significant gap in existing research that primarily focuses on single-person scenarios like dancing.

-  The KungFu-Fiesta (KFF) dataset, created using Unity, provides a rich and diverse collection of combat scenarios, enabling the model to learn from a variety of interactions and poses.

-  The paper includes thorough qualitative and quantitative evaluations, demonstrating the superiority of MagicFight over existing methods.

**Limitations:**

-  Given two reference IDs and the pose sequence, the proposed framework is able to generate a suitable video. Usually, how can we obtain the pose sequence for a video, especially in real-world scenarios rather than synthetic scenarios? How can we align the pose sequence with the reference IDs and is there a possibility that the pose sequence could be inconsistent with the described language or reference IDs?

-  Compared to generating a single person, generating interactions between two people focuses on the difficulty of creating decoupled actions due to potential mutual occlusions. I guess that the model could struggle with generating occluded parts during action sequences, especially when two characters overlap, which affects the realism and accuracy of the generated videos.

-  The integration of various modules and the use of a bespoke dataset might lead to increased computational complexity and demand, potentially limiting scalability. The authors could also show the count of parameters and inference time in the experimental section.

-  Some references related to action generations are missing [1,2,3], which can be added.

[1] Tm2d: Bimodality Driven 3d Dance Generation via Music-text Integration, in ICCV2023.

[2] Co-Speech Gesture Video Generation via Motion-Decoupled Diffusion Model, in CVPR2024.

[3] StructLDM: Structured Latent Diffusion for 3D Human Generation, in arxiv2024.

**Suitability:**

3

---

### Official Review · Reviewer_4RT5 · 2024-06-05

**Rating:** 4
**Confidence:** 2

**Summary:**

The paper focuses on text-to-video generation with two-person interactions. Thus, the paper proposes MagicFight and a bespoke dataset for conditioned video generation in the context of martial art combat. Experimental results showed the effectiveness of the proposed method.

**Strengths:**

1. The topic, i.e. multi-agent conditional video generation is more general with more applications, while remains less unexplored in the literature.
2. The proposed method seems reasonable and can achieve good performance.
3. The paper is well structured and easy to follow.

**Limitations:**

1. Although two-person fighting cases are not considered by previous methods, there already exist methods for multi-person conditional video generation, which is a more general task. What are the specific challenges of two-person fighting video generation? What specific designs are proposed to tackle the challenges?
2. How to alleviate identity confusion, model complex interactions between identities, and ensure consistency over time are the main challenges in multi-agent conditional video generation. Although the authors claim that the proposed method is specially crafted to overcome these burdens, how and why the method can alleviate these are not clearly stated.
3. Are there any anonymous links that can show the generated videos by the proposed method?

**Suitability:**

2

---

### Meta-Review · Area_Chair_ccXw · 2024-07-01

**Recommendation:** Accept (Poster)
**Confidence:** 4

**Metareview:**

The paper introduces "MagicFight," a framework designed for generating text-to-video content focusing on two-person martial arts interactions. This approach incorporates a specially designed dataset to address the unique challenges of accurately depicting complex interactions in martial arts combat. Experimental results support the effectiveness of MagicFight.

Strengths:

+ Relevance and Innovation: The focus on two-person interactions in video generation, specifically in martial arts, fills a gap in the current research landscape, which has predominantly concentrated on single-agent scenarios.

+ Structured Presentation: The paper is well-organized, making it accessible and easy to comprehend, which is beneficial for both seasoned researchers and those new to the field.

+ Effective Methodology: The proposed method not only addresses the generation of complex video content but does so with a level of performance that surpasses existing methods, as demonstrated through both qualitative and quantitative assessments.

Limitations:

+ Scope of Method Application: While the method excels in two-person scenarios, the paper lacks a detailed discussion on its applicability or adaptation to scenarios involving more than two individuals or less controlled environments.

+ Detail on Specific Challenges: The paper does not thoroughly explain how it resolves specific technical challenges unique to two-person interactions, such as identity preservation and interaction dynamics, which are critical for realistic video generation.

+ Availability of Demonstrative Materials: There is no direct provision of links or supplementary materials that allow for the independent evaluation of the generated videos, which could help in better assessing the visual quality and effectiveness of the proposed method.

The recommendation leans towards a borderline acceptance, indicating potential for impact but also highlighting areas where further detail and clarification could strengthen the overall contribution. The confidence in this recommendation is somewhat confident, reflecting the need for additional material to fully assess the method's robustness and generalizability.  It is highly recommended that the code be made publicly available.